# Addition of Combinedly Dehydrated Peach to the Cookies—Technological Quality Testing and Optimization

**DOI:** 10.3390/foods11091258

**Published:** 2022-04-27

**Authors:** Vladimir Filipović, Biljana Lončar, Jelena Filipović, Milica Nićetin, Violeta Knežević, Vanja Šeregelj, Milenko Košutić, Marija Bodroža Solarov

**Affiliations:** 1Faculty of Technology, University of Novi Sad, Bul cara Lazara 1, 21000 Novi Sad, Serbia; cbiljana@uns.ac.rs (B.L.); milican@uns.ac.rs (M.N.); ovioleta@uns.ac.rs (V.K.); vanjaseregelj@uns.ac.rs (V.Š.); 2Institute of Food Technology in Novi Sad, University of Novi Sad, Bul cara Lazara 1, 21000 Novi Sad, Serbia; jelena.filipovic@fins.uns.ac.rs (J.F.); milenko.kosutic@fins.uns.ac.rs (M.K.); marija.bodroza@fins.uns.ac.rs (M.B.S.)

**Keywords:** cookies, osmodehydration, lyophilization, peach, quality optimization

## Abstract

Peach dehydrated by a combined method of osmodehydration and lyophilization is characterized by upgraded dehydration effectiveness and enhanced chemical and mineral matter content, and as such, is an interesting material to be applied to the cookies’ formulation. Incorporation of this material requires testing and optimization of the addition level from the aspect of overall technological quality in order to obtain a new cookie product. Obtained cookie samples with different levels of dehydrated peach addition were subjected to the nutritive and technology quality parameters testing. Cookies’ chemical, mineral matter, and phenolic compounds content, the antioxidative activity of nutritive parameters, and the physical, technological, textural, colour, and sensory characteristics of technological parameters were investigated. Obtained results showed that the addition of especially higher levels of dehydrated peach enhanced all nutritive, while simultaneously decreased most of the technological quality parameters. The statistical method of Z-score analysis was used to calculate the optimal level of dehydrated peach addition to the cookie formulation for obtaining the highest nutritive enrichment without excessive technological quality deterioration. The optimal addition of osmodehydrated and lyophilized peach to the cookie formulation was determined to be 15%.

## 1. Introduction

Different combinations of various material dehydration methods have proven to provide numerous beneficial results. For example, a combination of the osmotic dehydration method with lyophilization can reduce the application volume of a high energy-demanding process that produces excellent product quality (lyophilization) via application of the osmotic dehydration process as pre-treatment, which is characterized by low energy requirements [1,2,3].

Providing the new-use value for food industry waste products and streams is a paramount task due to the current impact on the environment of the highly industrialized world. Numerous studies have investigated the use of sugar beet molasses, a by-product of the sugar industry, as an osmotic solution in dehydration of different animal and plant tissues [3,4,5,6,7].

Molasses is characterized by high dry matter content and rich nutritive composition, which provides the potential to be a successful osmotic solution, due to high driving force for the water removal during the osmotic dehydration process. Furthermore, its specific chemical composition, sensory properties, low cost, and reuse of by-products of different industry, positions molasses as a better choice than other conventional osmotic mediums [8].

Peaches are aromatic fruits with a specific, enjoyable, sweet taste, high organoleptic properties, and many commercial varieties [9,10,11]. Peach also has favorable nutritional content, is free of sodium, fat, and cholesterol, and has a rich content of vitamins A and C [12].

Previous research by Filipović et al. [3] showed that the combined dehydration method of peach was characterized by upgraded overall dehydration effectiveness, reduced time and energy consumption, and enhanced chemical and mineral matter content of dehydrated peach samples.

Cookies are a top-rated wheat-based product consumed worldwide due to their different flavour varieties, reasonable price, long shelf life, and readiness to eat. In effort of health improvement and increase in conscience regarding diet and health interaction, cookies’ formulation adjustment is needed in order to enchase their nutritive and functional characteristics [13], where integrating dehydrating peach in cookies formulation is an effort to achieve these goals.

Cookies are a complex system where every ingredient has an essential purpose, and every change in standard formulation usually leads to cookie dough changes that affect final product quality [14]; hence every addition of new raw material to cookie formulations requires precise testing and quality optimization.

In this research, peach dehydrated by combined osmotic dehydration and lyophilization method was added to standard sweet cookies in order to test and optimize the level of addition to cookies formulation from the aspect of overall technological quality.

## 2. Materials and Methods

### 2.1. Material

Fresh peaches (*Prunus persica*, var. *nucipersica*, Big top cultivar) were purchased at the local grocery store, had initial dry matter content of: 7.40 ± 0.08%, and were prepared for processing by washing with running tap water, drying with paper towels, and peeling and cutting into cubes approximately 1 cm × 1 cm × 1 cm.

Sugar beet molasses was obtained from sugar factory Crvenka, Serbia, had dry matter content of 85.04%, and was used as an osmotic solution in the osmotic dehydration process.

Material needed for cookie production was purchased in a local store and was as follows: white wheat flour, type T-400, moisture content of 14%, produced by “Danubius”, Novi Sad, Serbia; margarine produced by “AD Dijamant”, Zrenjanin, Serbia; sugar produced by “Šajkaška” Žabalj, Serbia; NaCl produced by “SO Produkt”, Stara Pazova; NaHCO_3_, produced by “Aleva”, Novi Kneževac, Serbia; and glucose, produced by BIO-UNA Novi Sad, Serbia.

### 2.2. Osmotic Dehydration

In the combined method of peaches dehydration, an osmotic dehydration process in molasses was applied to part of the peaches samples as a dehydration stage prior to lyophilization. The osmotic dehydration process was conducted in laboratory vessels under atmospheric pressure in a thermostat chamber (Memmert IN160, Schwabach, Germany) at a constant temperature of 20 °C for 5 h. To reduce excessive molasses’ dilution, the quantity of molasses used in the process was adjusted to a weight ratio of peach cubes samples to molasses of 1:5. Then, peach samples were immersed in molasses and stirred every 15 min for better homogenization of defunded water that originated from the dehydrating peach and molasses. After the 5 h process, peach samples were taken out from the molasses, lightly washed with running tap water, and gently blotted to remove excess surface water.

### 2.3. Lyophilization

The lyophilization process of the exact parameters was applied to previously osmotically dehydrated and fresh peach samples. Hence, one part of the peach samples was dehydrated in the combined dehydration process, while the other part of the peach samples was dehydrated in a one-stage lyophilization process.

Both fresh and osmotically dehydrated peach samples were, prior lyophilization process, frozen and stored at −30 °C. Frozen samples were placed in a freeze dryer (Christ ALPHA1-2 LDPLUS, Osterode am Harz, Germany) and lyophilization parameters were set to: pressure of 1.6 Pa, condenser temperature of −57 °C, and duration of 5 h.

Parameters of osmotic dehydration and lyophilization processes were selected based on defined optimal parameters in previous research [3].

### 2.4. Cookie Samples Production

Cookies samples with and without the addition of lyophilized and osmodehydrated and lyophilized peach production, which comprised production operations of dough mixing, processing, and baking, were conducted in a pilot plant for bakery products of Institute of Food Technology in Novi Sad, Serbia, by AACC method 10–50 D [15]. Experimental plan design that relies on different cookie dough formulation is presented in Table 1.

In the effort of obtaining the same moisture content and comparability of different cookie dough while applying material of different moisture content (lyophilized (L) peach was characterized by a dry matter content of 20.14%, while osmodehydrated and lyophilized (OL) peach sample had 70.10% of dry matter content [3]), the quantity of distilled water addition was adjusted to every cookie sample with dehydrated peach formulation, providing the same level of dough moisture content through the investigated cookie samples. The technological possibility of obtaining undeteriorated dough structure determined levels of dehydrated peach addition to the cookies’ formulation. The maximal level of L peach addition had proven to be 5% of peach dry matter content based on flour dry matter content. In contrast, in the case of OL peach, the maximal level of addition was 25%, due to the significantly higher dry matter content of peach dehydrated by the combined method (OL). Peach samples were cut into small pieces of approximate dimensions of 0.5 cm × 0.5 cm × 0.5 cm, in order to better homogenize with the rest of the dough components.

Dough mixing was done by multi-stage procedure, where weighted ingredients of margarine, sugar, NaCl, NaHCO_3_, and glycose were placed in a mixer bowl and mixed for 3 min at the lowest mixing speed. In the second mixing stage, distilled water was added and mixing was continued for another 1 min at the lowest speed, after which mixing was continued at medium speed for another 1 min. At the third mixing stage, total quantities of flour and dehydrated (L or OL) peach were added (Table 1) and mixed at the lowest speed for 2 min. Every 30 s, dough residues were scraped from the mixing bowl walls, and after the mixing stage competition, homogenous dough without lumps was obtained.

The obtained dough was rounded by hand and placed in PVC bags and in the refrigerator at 8 °C for 30 min for resting.

Rested dough was laminated on a laminating device (Mignon, Rimini, Italy) by gradual dough thinning via passing between a set of rollers with decreasing clearance (14 mm, 10 mm, and 7 mm).

Dough shaping was done by ϕ = 60 mm mold pressing, providing samples of the uniform weight of 35 g.

Shaped dough baking was done in the floor oven (MIWE Gusto CS, Arnstein, Germany) at the temperature of 205 °C for 10 min.

Baked biscuits were left for cooling and resting at controlled ambient conditions (23 °C and relative humidity of 60%) for 30 min, packed in polypropylene bags, and stored before further testing.

### 2.5. Methods of Cookie Chemical Composition Determination

Proximate chemical composition of cookies was performed according to AOAC standard methods [16]: protein content-method No. 950.36, total carbohydrates content–method No. 2020.07 starch content-method No. 996.11, reducing sugars -method No. 80-68, fat content-method No. 935.38, cellulose content-method No. 973.18, ash content-method No. 930.22, and water content-method No. 926.5. Each measurement was performed in three repetitions.

### 2.6. Methods of Cookie Mineral Matter Composition Determination

The mineral content of potassium (K), calcium (Ca), magnesium (Mg), and iron (Fe) of the cookies was determined according to the standard methods of AOAC [16]. Minerals were determined by atomic absorption spectrophotometry (method No. 984.27) on a Varian Spectra AA 10 (Varian Techtron Pty Ltd., Mulgvare Victoria, Australia). Each measurement was performed in three replications.

### 2.7. Methods of Total Cookie Carotenoids, Polyphenol Content and Antioxidant Activity Determination

For determination of carotenoids content, 2 mL hexane was mixed with 500 mg of ground cookie sample on Vortex for 2 min, then centrifuged at 12,000 rpm for 3 min. The liquid part was collected and filtered through a 0.45 μm filter. The content of carotenoids in cookies extracts was analyzed spectrophotometrically by Nagata and Yamashita [17], using extracting solvent as the blank. Results were expressed as mg of β-carotene equivalents per 100 g sample (mg β-car/100 g).

To determine total polyphenols and antioxidant activity, 2.5 mL ethanol, acetic acid, and water (50:8:42) were mixed with 500 mg of ground cookie sample on Vortex for 2 min, then centrifuged at 12,000 rpm for 3 min. The liquid part was collected and filtered through a 0.45 μm filter. Total polyphenol contents in cookie extracts were determined by the Folin-Ciocalteau method adapted to microscale [18]. Results were expressed as mg gallic acid equivalents per 100 g cookie (mg GAE/100 g).

The antioxidant activity was assessed following three different methods: 2,2-diphenyl-1-picrylhydrazyl (DPPH) according to Tumbas Šaponjac et al. [19], 2,2′-azino-bis-3-ethylbenzothiazoline-6-sulphonic acid (ABTS) as described by Aborus et al. [20], and reducing power (RP) as outlined by Oyaizu [21]. Results were expressed as μmol of Trolox equivalents per 100 g cookie (μmol TE/100 g).

### 2.8. Methods of Cookie Technological Quality Analysis

The cookies’ technological quality parameters were determined by AACC 10–50D [15] method.

Baking weight loss (BWL) was determined by measuring the weight of cookies before and after the baking stage of production and applying the following equation:(1)$$\mathrm{BWL}\;\left(\%\right)\;=\;\frac{\left(m_{0}\;-\;m_{t}\right)}{m_{0}}\;100$$
where, m_0_ is cookies weight before baking (g) and m_t_ is cookies weight after baking (g).

Cookie dimensions measurements were done after 30 min of the cooling period. Measurements of cookie diameter in the lamination direction (length–L), cookie diameter normal to the lamination direction (width–W), and cookie thickness (T) were performed.

Average cookie diameter (R) was determined by the lowest W and highest L.

T was measured by stacking six cookies and measuring their total height. After the first measurement, six cookies were rearranged in the column and their height was measured again. Finally, the median value of these measurements was divided by the number of cookies (6) to calculate the median value of cookies T.

The widening factor (R/T) was determined by the ratio of median values of R and T, which indicates cookies’ shape deformation during baking.

### 2.9. Methods of Cookie Texture Instrumental Analysis

Cookie texture characteristics were determined on a texture analyzer TA-XT2 Texture Analyser (Stable Micro System, Godalming, UK) equipped with a 25 kg load cell and Knife Edge with Shotted Insert HPD/bs tools. Measurements were done by using compression mode at the crosshead speed of 1 mm/s prior, 3 mm/s during, and 10 mm/s after the analysis. By application of computer software Exponent Stable MicroSystems, version 6.0, maximum force (*n*) and distance at break were registered in the function of time, and represented indicators of cookies’ hardness. Cookies’ textural characteristics measurements were performed in six repetitions, at each cookie batch, 24 h after the baking, at a temperature of 25 °C and with dimensions of 50 mm × 50 mm.

### 2.10. Methods of Cookie Colour Instrumental Analysis

Colour parameters of cookie sample surface were determined in six replications, 24 h after baking, using Chroma meter (CR-400, Konica, Minolta, Tokyo, Japan) tri-stimulus colorimeter (contact surface diameter: 8 mm). Prior to samples measurement, calibration was done using the white colour standard. Colour parameters results were presented according to CIElab colour system, where coordinates are defined as following: L*—brightness (from 0 (black) to 100 (white)), a*—greenness/redness (from −a* (green) to +a* (red)), b*—blueness/yellowness (from −b* (blue) to +b* (yellow)) [22,23].

Colour variation between control cookie sample and cookie samples with dehydrated peach addition (ΔE), was determined by the following equation:(2)$$∆E\;=\;\sqrt{∆L^{2}\;+\;∆a^{2}\;+\;∆b^{2}}$$
where are:
ΔL*—difference in L* parameter between control and cookie sample with dehydrated peach addition,Δa*—difference in a* parameter between control and cookie sample with dehydrated peach addition,Δb*—difference in b* parameter between control and cookie sample with dehydrated peach addition.


### 2.11. Methods of Cookie Descriptive Sensory Analysis

A panel of eight trained evaluators (six women and two men) of the Sensory and Technical Analysis Department of the accredited laboratory of the Institute of Food Technology, Novi Sad, Serbia, conducted a descriptive sensory evaluation to obtain cookies’ sensory samples profile. A panel of trained evaluators was formed according to appropriate standards: ISO 6658:2017 [24], ISO 8586:2012 [25], ISO 3972:2011 [26], ISO 5495:2005 [27], ISO 11037:2011 [28], and ISO 11036:2020 [29].

Descriptor selection for cookie samples sensory profiling was previously conducted by the evaluators’ panel leader and further adjusted with the rest of the evaluators to better define cookies’ sensory profiles. The final list comprised of six descriptors, where two descriptors characterized cookie appearance (colour intensity (CI) and surface appearance (SA)), one descriptor characterized deviation from standard taste (T), one descriptor characterized deviation from the standard smell (S), and two descriptors were used for textural propertied definition (hardness (sensory hardness–SH) and fracturability (F)). The seven-point scale measured the intensity of each descriptor, where 1 was described as the lowest intensity and 7 as the highest intensity, [30], except for descriptors for T, S, SH, and F, where optimal descriptor values were set to value 4, and negative or positive deviations from this value were characterized by different cookie samples sensory attributes (Appendix A).

Cookie sample sensory evaluation was performed 24 h after baking in the laboratory for sensory analysis of Institute of Food Technology, Novi Sad, Serbia, designed by standard ISO 8589:2007 [31]. Cookie samples were presented to the evaluators on white plastic plates, coded by random three-digit codes from the table of random numbers, and every evaluator tested five samples per session [32,33]. After every cookie samples’ testing, evaluators washed their mouths with water. Used descriptive sensory analysis form for cookies samples testing is presented in Appendix A.

### 2.12. Methods of Statistical Analysis

#### 2.12.1. Correlation Analysis

The colour plot diagram for mean values of all nutritive and technological quality responses (32 in total) of cookies with and without the addition of liophilized and osmodehydrated and lyophilized peach was calculated and plotted using R software v.4.0.3 (64-bit version). The corrplot instruction was applied, with the “circle” method, upper type enabled, as a graphical tool to represent the correlation between tested responses of observed samples.

#### 2.12.2. Principle Component Analysis

Principal component analysis (PCA) was applied as the pattern recognition technique for using assay descriptors to characterize and differentiate various analyzed samples and their responses.

#### 2.12.3. Analysis of Variance

Statistical differences were determined by an analysis of variance (ANOVA), with mean separations performed by the Tukey HSD test.

ANOVA and PCA analysis were performed using STATISTICA 12.0 software (2013), (StatSoft Europe, Hamburg, Germany).

#### 2.12.4. Z-Score Analysis

The Z-Score analysis applies min-max normalization to the different cookies with dehydrated peach quality characteristics, transforming these response values from their original unit system to a new dimensionless unit system, where these different responses are comparable and further mathematical calculations are applied [34,35].

The maximum value of normalized total Z-score presents the optimum value of all combined segment Z-scores, including all analyzed responses, indicating the optimum total quality of cookies’ samples. The following equations show the calculation of individual segment Z-scores:

Chemical composition segment Z-score:(3)$$S_{1i}\;=\;\frac{\sum_{k\;=\;1}^{5}\left(\frac{x_{ki}\;-\;x_{k\;min}}{x_{k\;max}\;-\;x_{k\;min}}\right)\;+\;\sum_{j\;=\;1}^{2}\left(1\;-\;\frac{x_{ji}\;-\;x_{j\;min}}{x_{j\;max}\;-\;x_{j\;min}}\right)}{7}$$
where *x_k_* are: protein, total carbohydrates, sugar, cellulose, and ash content; and *x_j_* are: starch and fat content.

Mineral matter composition segment score:(4)$$S_{2i}\;=\;\frac{\sum_{l\;=\;1}^{4}\left(\frac{x_{li}\;-\;x_{l\;min}}{x_{l\;max}\;-\;x_{l\;min}}\right)}{4}$$
where *x_l_* are: K, Ca, Mg, and Fe.

Content of phenolic compounds and antioxidative activity segment score:(5)$$S_{3i}\;=\;\frac{\sum_{m\;=\;1}^{5}\left(\frac{x_{mi}\;-\;x_{m\;min}}{x_{m\;max}\;-\;x_{m\;min}}\right)}{5}$$
where *x_m_* are: total phenolic content, total carotenoid content, antioxidative activity by DPPH method, reduction potential, and antioxidative activity by ABTS method.

Technological quality responses segment score:(6)$$S_{4i}\;=\;\frac{\sum_{n\;=\;1}^{2}\left(\frac{x_{ni}\;-\;x_{n\;min}}{x_{n\;max}\;-\;x_{n\;min}}\right)\;+\;\sum_{o\;=\;1}^{4}\left(1\;-\;\frac{x_{oi}\;-\;x_{o\;min}}{x_{o\;max}\;-\;x_{o\;min}}\right)}{6}$$
where *x_n_* are: moisture content and thickness; and *x_o_* are baking weight loss, dimater, R/T ratio, and hardness.

Instrumental colour responses segment score:(7)$$S_{5i}\;=\;\frac{\sum_{p\;=\;1}^{3}\left(\frac{x_{pi}\;-\;x_{p\;min}}{x_{p\;max}\;-\;x_{p\;min}}\right)\;+\;\left(1\;-\;\frac{x_{ri}\;-\;x_{r\;min}}{x_{r\;max}\;-\;x_{r\;min}}\right)}{4}$$
where *x_p_* are: L*, a*, and b*; and *x_r_* is ΔE.

Descriptive sensory analysis segment score:(8)$$S_{6i}\;=\;\frac{\sum_{q\;=\;1}^{5}\left(\frac{x_{qi}\;-\;x_{q\;min}}{x_{q\;max}\;-\;x_{q\;min}}\right)\;+\;\left(1\;-\;\frac{x_{si}\;-\;x_{s\;min}}{x_{s\;max}\;-\;x_{s\;min}}\right)}{6}$$
where *x_q_* are: surface appearance, taste, smell, hardness, and fracturability; and *x_s_* is colour intensity.

Total quality characteristics Z-score:(9)$$S_{i}\;=\;0.15\;\cdot \;S_{1}{}_{i}\;+\;0.15\;\cdot \;S_{2}{}_{i}\;+\;0.3\;\cdot \;S_{3}{}_{i}\;+\;0.15\;\cdot \;S_{4}{}_{i}\;+\;0.15\;\cdot \;S_{5}{}_{i}\;+\;0.1\;\cdot \;S_{6}{}_{i}$$
where cookies’ nutritive quality characteristics Z-scores values represent 60%, while techological quality characteristics Z-scores values represent 40% of total Z-score, or total quality.
max [*S_i_*]→optimum(10)

Z-score calculation was performed using Microsoft Excel ver. 2016.

## 3. Results and Discussion

### 3.1. Chemical and Mineral Matter Content of Cookies

Nutritive characteristic changes of cookies that occur with the addition of differently dehydrated (L or OL) peach were investigated via analysis of their chemical, mineral matter, and phenolic compounds compositions, and antioxidative activity.

Table 2 shows the results of the chemical and mineral matter content of different cookie samples, with the addition of different quantities of L and OL peach, together with the control sample (without the addition of peach).

The addition of L and OL peach to the cookie formulation led to increased protein, carbohydrate, sugar, cellulose, and ash content (Table 2). In all tested samples, the increase was only statistically significant in cookie samples with OL peach addition of at least 10%. Explanation of these cookie samples’ chemical content response increase can be provided from chemical content of added dehydrated peaches [36] to the cookie formulation, which supplemented these tested responses. Starch and fat cookie content decreased with the addition of dehydrated (L or OL) peaches, where for the statistically significant decrease, the quantity of addition had to be, again, at least 10%. This is due to the fact that the addition of dehydrated peach, a material of low fat and starch content, substituted an equivalent amount of flour. There was no statistically significant difference between chemical content responses of cookie samples with added L and OL peach. However, protein, starch, cellulose, and ash levels were higher in cookie samples with added OL peach of the same addition amount. Maximal values of protein, total carbohydrates, sugars, cellulose, ash, and all tested mineral matter contents were found in sample 9 (cookie sample with the highest quantity addition of OL peach), while the lowest contents of starch and fat characterized this sample. On the other hand, the highest starch and fat content was detected in sample 1 (control cookie sample, without peach addition).

All mineral matter content responses of cookies with OL peach addition were statistically significantly higher than samples 1 to 3 (control sample and cookies with L peach addition). It indicates that molasses’ rich mineral matter composition [37], via solid gain in the osmotic dehydration process stage [38], statistically significantly enriched mineral matter content of cookies (Table 2). The direct contribution of the osmodehydration stage in the combined dehydration process (OL) of peach on cookies’ mineral matter content can be seen by comparing cookie samples 5 and 3, where the increase in K, Ca, Mg, and Fe, was determined to be: 16.21%, 16.99%, 15.16%, and 6.35%, respectively.

### 3.2. Total Carotenoid and Polyphenol Contents and Antioxidative Activity of Cookies

Table 3 shows the results of total carotenoid and polyphenol contents and antioxidative activity responses of nine different cookie samples, with the addition of different quantities of L and OL peach.

The analysis of total carotenoid content showed that minimal OL peach addition of 10% to the cookies formulation was needed to detect this response. Higher additions had led to a statistically significant increase in total carotenoids of cookies supplemented with peach dehydrated by combined method (OL). The peach was reported to contain high levels of total carotenoids, particularly zeaxanthin, lutein, β-carotene, and β-cryptoxanthin [39]. Among these carotenoids, β-carotene and β-cryptoxanthin are provitamin A compounds, which play an important role in immune system function. The addition of β-carotene rich material to wheat-based products also produces an increase in β-carotene content of the final products, as reported by other authors [40].

The results of all total polyphenol content and antioxidant activity responses of different cookie samples indicate a trend of statistically significant effects of dehydrated peach addition to cookie formulation (Table 3). Similarly, several authors found that enrichment of cookies with polyphenol antioxidants from different fruits and vegetables led to an improvement of antioxidant properties [13,18]. Furthermore, introducing wheat malt flour–of 3 to 4 days of germination–to the cookies formulation also increases phenolic content [41] by a similar mechanism as dehydrated peach addition.

The results of total polyphenol content showed that dehydrated peach addition had led to the statistically significant increase in cookies’ total polyphenols, even at the lowest levels of addition (Table 3). Comparison of the cookie samples with the same level of peach addition but dehydrated by different methods (samples 2 and 4; 3 and 5) indicates no statistically significant increase. This comparison indicates that the main source of the cookies’ total polyphenol content was peach. In addition, molasses is known for its high total polyphenol content [42], hence via incorporation to the peach dehydrated by combined method (OL) it had influenced an increase in this response of cookie samples. This is especially noticeable in cookie samples with high levels of OL peach addition.

In practice, a single assay method is not sufficient for in vitro assessment of antioxidant activity of endogenous phytochemicals; different assays vary in terms of mechanisms and experimental conditions.

In addition, antioxidant molecules differ in polarities, thus they can act as a radical scavenger by electron-donating mechanism or by hydrogen-donating mechanism. The antioxidant activity of cookies rich in both hydrophilic and lipophilic molecules was investigated by three methods, i.e., by measuring scavenger activity on DPPH and ABTS radicals, and reducing power.

The results of the antioxidant activity of cookie samples, determined by the DPPH method, indicate the statistically significant effect of dehydrated peaches addition to cookie formulation, even at the lowest levels (2.5% of addition) (Table 3). As previously mentioned, peaches are characterized by high antioxidant activity, also confirmed by the DPPH method [43]. On the other hand, the contribution of molasses present in dehydrated peaches, produced in the combined method (OL), to the cookies’ overall antioxidant activity, measured by the DPPH method, was not statistically significant. However, it contributed to the 20% and 22% increase in this response in samples with 2.5% and 5% additions, respectively.

The results of reducing power show precisely the same trends of statistically significant effects of the dehydrated (L or OL) peaches’ addition to cookie formulation, as described in the case of antioxidant activity, determined by the DPPH method (Table 3).

Antioxidant activity of cookie samples, determined by the ABTS method, was statistically significantly increased even with the lowest level of addition of lyophilized peach, indicating the high antioxidant potential of peach, which is also determined by previous analysis of this research and by other authors [43] (Table 3). Furthermore, there was also a statistically significant effect of peaches with the combined dehydration method (OL), where molasses’ high antioxidant activity [42] influenced the overall increase in cookies’, antioxidant activity, determined by the ABTS method: up to 2.08 and 2.29 times in cookies samples with addition of 2.5% and 5%, respectively, in comparison to the cookies with L peach addition.

Higher levels of OL peach addition to the cookie’s formulation (from 10% to 25%) had provided far superior nutritive responses in comparison to cookie samples 1 to 5, due to the combined effect of higher added quantities of molasses and peach dry matter to the cookies’ formulations (Table 2 and Table 3).

Maximal values of all tested total carotenoid and polyphenol contents and antioxidative activity responses were observed in cookie sample 9.

The effect of dehydrated peach addition on the cookie formulations’ nutritive characteristics are in accordance with the research of Salehi and Aghajanzadeh [44], where it is also reported that nutritional values of prepared cakes with different fruits powder addition significantly increased.

### 3.3. Technological Quality Parameters of Cookies

Results that describe the technological characteristics of cookies with the addition of L and OL peach are presented in Table 4, Table 5 and Table 6.

Cookies’ technological quality depends on raw materials used for formulation: flour quality, fat addition, sugar, water, and other materials. With the mechanical energy application, raw material components interact to create a dough that produces a final product of specific physical, chemical, and sensory characteristics after the baking stage. In the baking stage, dough changes occur, colour, taste, and smell are formed, moisture decreases, and cookie dimensions change [45].

With the addition of different quantities of L and OL peach, technological quality responses of cookies are presented in Table 4, from where the statistically significant influence of dehydrated peach addition (regardless of dehydration method) on cookies’ moisture content can be seen. All cookie samples with added dehydrated peach had statistically significantly lower moisture content than the control sample (without dehydrated peach addition). Adjusted cookie dough formulation by an experimental plan, that targeted the same quantity of cookie dough moisture level, regardless of moisture source (total quantity of water added directly to flour in case of sample 1, or reduced quantity of water for the amount existing in dehydrated (L or OL) peach added in all other samples) is maybe the explanation to these cookie moisture content results. Water added directly to the flour in total quantity, according to standard cookie formulation [15], probably had a better effect on dough formation and dough moisture distribution than water added to flour in reduced quantity, where part of the water was added via (partly) dehydrated peach. The dough of cookies with dehydrated peach probably exerted less uniform moisture distribution, due to more moisture retention in dehydrated peach material and less flour hydration. During the cookies’ baking stage, water retained in dehydrated peach material was probably more available to evaporation, consequently lowering the moisture content of baked cookies.

Other researchers [40] also reported a decrease in moisture content of final wheat products with the addition of dehydrated fruit products.

There was also notable statistically significantly lower moisture content in cookies with L peach than in cookies with OL peach, at the same addition level. The same mechanism of water addition to the cookie dough can be offered as an explanation for these results, since there was a significant difference between the moisture content of L peach and OL peach, which caused adjustment to different water quantities in addition to dough formulation. With the increase in dehydrated peach quantity addition, moisture content statistically significantly decreased.

The baking weight reduction response results follow the same trends of a statistically significant effect of dehydrated peaches addition to cookies formulation, as described for moisture content (Table 4), except the trends are in negative correlation. Therefore, the same mechanisms of water addition to the cookie dough via different sources and its effect on dough and cookie characteristics, as in the case of cookies’ moisture content, can be proposed as an explanation for the baking weight reduction response increase with the quantity of dehydrated peach addition.

Moisture content and baking weight reduction are important technological quality parameters since they indicate texture and yield of the final product.

Cookie dimensions are significant properties of baked products quality control, and can also be used for defining the effect of different material addition on products’ technological characteristics [45].

The results of cookie samples’ diameters (Table 4) indicated no statistically significant difference between all cookie samples, except for sample 9, where the highest level of peach dehydrated by combined method (OL) addition statistically significantly affected cookie diameter increase. The peaches’ different dehydration methods did not significantly affect cookie diameters.

The thickness of the cookie samples was statistically significantly decreased, even with the lowest levels of dehydrated peaches additions (Table 4). The type of peaches dehydration method affected the thickness of the cookie samples, where statistically significant thickness values were determined in cookie samples with peach dehydrated by OL method addition. This observation can be explained by the same mechanism which is proposed to explain higher levels of moisture content of cookie samples with the addition of peaches dehydrated by the OL method.

Higher levels of dehydrated peach addition (samples 6–9) influenced a statistically significant decrease in cookie sample thickness. Cookie sample thickness is the result of a balance between the setting of the cookie structure by thermal denaturation of the gluten network and the expansion of the dough by the action of the aerating agents and the steam [45], hence any added material to the cookie formulation in the quantity that can disturb cookie structure will likely cause cookie samples thickness reduction. In addition, as investigated by other authors [40,44], replacement of wheat flour with different cellulose-rich material, causes a reduction in dough gluten content, producing lower final product volume, hence reducing thickness.

The results of the R/T ratio (Table 4) showed that supplementation of dehydrated (L or OL) peach to the cookie formulation, even at the lowest levels of addition, caused a statistically significant increase in this response, which indicates deformation of the cookie samples’ shape. Analysis of the effect of dehydration type on samples’ R/T ratio showed that cookie samples with peach dehydrated by the OL method were characterized by a statistically significantly lower R/T ratio, hence lower shape deformation, at the same level of dehydrated peaches addition (10.33% and 23.65% lower deformation at the addition level of 2.5% and 5%, respectively). Further increase in addition of peach dehydrated by OL method to the cookies’ formulation (samples 6–9) statistically significantly increased samples’ shape deformation. Since this parameter is only a mathematical combination of two previously discussed responses (cookie samples’ diameter and height), the same explanation of the acting mechanisms of the effect of dehydrated peaches on cookies’ structure can be proposed.

Cookie hardness, results of which are shown in Table 4, represents the necessary force at which total break of the structure occurs [13], and from these results it can be seen that dehydrated peach addition had led to a statistically significant increase in cookie hardness.

These results can be correlated to cookies’ moisture content results, where the same proposed mechanism of water distribution and its evaporation during all phases of production, can be used to discuss the cookie hardness results. Cookie samples 4 and 5 had a statistically significantly lower hardness than samples 2 and 3, with corresponding dehydrated peach addition, indicating that peaches’ OL method had a statistically significant effect on lowering cookies’ hardness in comparison to L peaches (the results were 2.06 and 1.71 lower for the corresponding samples with 2.5% and 5% of dehydrated peaches addition, respectively). Increasing the quantity of dehydrated peaches addition (samples 6–9) led to a statistically significant increase in cookies’ hardness. These findings are correlated to the research of Salehi and Aghajanzadeh [44], where the addition of different fruit powders to the batter formulation produced firmer texture, or higher hardness of final products, and also to the findings of Shabnam et al. [46], where texture quality parameters of wheat cookies decreased with an increase in peach powder level addition.

With the exclusion of sample 1 since it was used as a control sample (cookie without dehydrated peach addition), the highest values of moisture content and thickness, and lowest values of baking weight loss, diameter, R/T, and hardness, as the preferable technological quality parameters, were determined in samples: 5, 4, 4, 2, 4, and 4, respectively.

### 3.4. Instrumental Colour Responses of Cookies

Colour is a significant element for consumers’ initial acceptability of cookie products [45]. Table 5 shows four instrumental colour responses of different cookie samples, adding different of L and OL peach quantities.

The addition of dehydrated peach to the cookies’ formulation had led to a statistically significant reduction in cookie surface lightness, while peach previously osmodehydrated in molasses, further statistically significantly decreased cookie lightness values. Other researchers [40,44] also reported that replacing flour with dried fruits produces the darker colour of final products.

The results show a statistically significant increase in cookie samples’ a* values, or increase in redness, with the addition of dehydrated (L or OL) peach to the cookies formulation, and also statistically significant increase with the addition of peach dehydrated in the OL process. Values of b* statistically significantly decreased with the addition of dehydrated peach, especially if added dehydrated peach was previously subjected to osmotic treatment, indicated by reducing yellow colour tone in these cookie samples. Colour difference of cookies with peach addition compared to control sample 1 statistically significantly increased with the addition of higher quantities of dehydrated peach addition, and also by using peach dehydrated by combined (OL) method.

Profound cookies colour change in samples with added peach dehydrated by the OL method can be attributed to the molasses’ impact on overall cookie colour appearance, since molasses, well known for its dark colour [37], colours dehydrated peach via solid gain [38], and also catalyzes developing Maillard reactions and caramelization, which affect overall cookies colour change.

The highest values of L* and ΔE responses were obtained for cookie samples 1 and 9, respectively, while the highest values of a* and b*, indicating the most red and yellow colour, were determined for samples 9 and 1, respectively.

### 3.5. Descriptive Sensory Analysis of Cookies

In Table 6, six descriptive sensory analysis responses of cookies with and without the addition of L and OL peach are shown.

Colour intensity followed the same statistically significant trend as in the case of instrumental colour measurement response of L * (higher quantity of addition and peach dehydrated by OL method produced statistically significantly higher intensity of cookies colour). Cookie samples’ surface appearance statistically significantly deteriorated with the addition of higher quantities of dehydrated peach, while the method of dehydration also affected surface appearance, where peach dehydrated by OL method addition led to higher cookie surface appearance deterioration compared to the L peach addition, at the same addition level.

The cookies’ sensory taste analysis showed that adding dehydrated peach (regardless of the dehydration method) to the cookie formulation, up to the level of addition of 10%, positively affected the cookies’ taste, providing a favorable peach note to the overall flavor. However, an increase in peach dehydrated by combined method (OL) addition, in quantities over 10% (cookie samples 7–9), statistically significantly decreased overall cookies’ taste, expressing a molasses note in the flavor.

Sensory smell analysis of the cookies shows that adding L peach to the formulation enhanced cookies’ smell by introducing peach notes to the overall scent. However, the addition of higher levels of peach, dehydrated by OL method, to the formulation statistically significantly decreased the overall cookies’ smell, expressing a molasses note in the cookies’ smell.

The cookies’ sensory hardness analysis showed that the addition of dehydrated peach statistically significantly increased hardness (lower descriptor values indicating higher sensory hardness), and that increased levels of dehydrated peach addition further statistically significantly increased hardness. Furthermore, the effect of the peach dehydration method can be seen by comparison of the cookie samples 3 and 5, where statistically significantly higher sensory hardness was documented for samples with the addition of peaches dehydrated by the OL method. These results of sensory hardness analysis are in accordance with hardness results obtained by instrumental analysis, Table 4.

The cookies’ sensory fracturability analysis showed that the addition of lyophilized peach increased cookies’ fracturability along with the increase in the addition level. The addition of OL peach led to higher crumbliness of cookies samples, especially at higher addition levels (15% and higher).

Presented results indicate that minimal results for colour intensity, and maximal results for all other five responses (surface appearance, taste, smell, sensory hardness, breakability) as the preferable sensory responses, with the exclusion of sample 1 as a control sample, were determined for samples 2, 4 and 3, respectively.

Sensory analysis conducted by the other researchers [44,46] showed that the acceptability of wheat products with dehydrated fruit products directly depends on the amount of added amounts of dehydrated products, hence strict optimization and control of the addition of these materials are needed.

### 3.6. Results of the Statistical Analysis

#### 3.6.1. Results of the Correlation Analysis

Figure 1 shows a colour correlation diagram between all 32 responses of nutritive and technological quality characteristics of tested cookies. Values of correlation coefficients between two tested responses are visually presented by colour (blue for positive and red for negative correlation) and the size of the circles.

The results of correlation analysis show a high level of positive correlation between the following responses: proteins, carbohydrates, sugar, cellulose, and ash of chemical content; all mineral matter content responses; all phenolic compounds and antioxidative activity responses; cookie diameter, R/T ratio and hardness of technological quality responses; a* and ΔE of instrumental colour responses; and colour intensity of descriptive sensory analysis. Previously stated responses were characterized by a high level of negative correlation with the following responses: starch and fat of chemical content; moisture and cookies height of technological quality responses; L* and b* of instrumental colour responses; and surface appearance, taste, smell, sensory hardness, and fracturability of descriptive sensory analysis. These results confirm previously individually discussed responses where the addition of dehydrated peach (especially by combined method) to the cookie formulation positively affected most of the nutritive quality responses, while technological responses were mostly negatively correlated to these nutritive responses.

The highest positive correlations were found to be between all responses of nutritive quality characteristics (protein, total carbohydrates, sugar, cellulose, and ash content of chemical content responses to mineral matter content and phenolic compounds content).

#### 3.6.2. Results of the PCA

PCA was applied to detect structure in the correlation between numerous experimentally detected responses and different tested samples that give complementary information [47].

Figure 2 presents the PCA results, where, for visualization of the trends in shown data and discriminating efficiency of the used descriptors, a scatter plot of samples was produced, presenting the first two principal components from PCA of the data matrix, first principal component at x-axis and the second at the y-axis.

A neat separation of nine tested cookie samples according to different quality responses can be seen from the presented scatter plot. The position of the samples in the figure was primarily influenced by the amount of dehydrated peach addition to the cookies’ formulation (with the increasing quantity of peach addition, and the location of the cookie samples moved from positive to negative first principle component values). Type of dehydration method influenced cookie samples’ position along with second principal component, where samples with L peach addition were positioned at higher positive values of second principle component, while samples with peach dehydrated by the OL method addition were located at negative second principal component values. Control cookie sample 1 was characterized by high values of moisture, taste, and thickness; samples 2 and 3 were characterized by high values of taste, smell, and fracturability; samples 4 to 6 were located in the center of the diagram, indicating medium values of all tested responses; while samples 7 to 9 were characterized by high values of following responses: most of the chemical content responses; all mineral matter content responses; all phenolic content and antioxidative activity responses; cookie diameter; a* of instrumental colour responses; and colour intensity of descriptive sensory analysis. Quality results showed that the first two PCs accounted for 94.76% of the total variance and could be considered sufficient for data representation.

The contribution of the responses to the F1 was almost equally distributed between chemical, mineral matter, phenolic content, antioxidative activity, instrumental colour, and some of the descriptive sensory analysis responses. In contrast, the contribution of the most significant responses to the F2 was 17.09%, 14.91%, 13.64%, and 12.05% for baking weight loss, hardness, moisture, and height, respectively.

#### 3.6.3. Results of the Z-Score Analysis

Z-score analysis was applied to identify optimal cookie samples formulation from the aspect of all 32 investigated nutritive and technological quality responses. In Table 7, the results of the Z-score analysis of cookies, with and without the addition of L and OL peach, are shown. The presented results show segment Z-score S1–S6, which correspond to Z-score results of chemical, mineral matter, and phenolic compounds content, technological quality parameters, instrumental colour, and descriptive sensory analysis responses, respectively.

The presented results show that the addition of peach dehydrated by the OL method led to the increase in segment Z-score values for all nutritive cookie characteristics: chemical, mineral matter and phenolic content, and antioxidative activity. Segment Z-score values for technological cookie characteristics–technological quality, instrumental colour and descriptive sensory analysis–declined with the addition of dehydrated peach to the cookies’ formulation, especially with the addition of peach dehydrated by the OL method.

Maximal values of S1–S3 were recorded for sample 9, while values of S4–S6 were recorded for samples 4, 2, and 2, respectively, with the exclusion of control sample 1.

Total Z-score values mathematically combine all segment Z-scores and indicate the optimal combination of all tested cookies’ nutritive and technological responses. The addition of dehydrated peach by OL method to the cookie formulation produced an optimal combination of tested quality characteristics. Samples with dehydrated peach by combined method addition had 26.32% and 45.15% higher total Z-score values than samples with L peach addition at the addition levels of 2.5% and 5%, respectively. The optimal addition of OL peach to the cookies formulation was determined to be in the quantity of 15% (sample 7), excluding control sample 1.

## 4. Conclusions

From the presented results, it can be concluded that different dehydration methods had statistically significantly different impacts on the nutritive characteristics of cookies, where molasses application in the osmotic treatment stage of the combined dehydration method had a positive effect on cookies with dehydrated peaches’ overall nutritive composition. Cookies with 10% of combinedly dehydrated peach addition nutritive quality indicates a significant increase in chemical, mineral, and phenol content, and antioxidative activity. Peaches’ combined dehydration method positively influenced certain cookies’ technological quality parameters (BWL, R/T, hardness) in comparison to cookies with the same amount of L peaches addition. The results of the descriptive sensory analysis showed that with the addition of up to 10% of dehydrated peaches to the cookie formulation, there was a positive effect on all sensory responses, providing a favorable peach note to overall taste and flavor. Correlation analysis results confirmed the positive effect of dehydrated peach addition to the cookies formulation on nutritive characteristics, while technological quality parameters were mostly negatively correlated. The results of the Z-score analysis indicated an optimal level of dehydrated peach addition regarding overall cookies quality (quantity of 15% of addition), proposing a new type of nutritively enriched cookie product as the main results of this research.

## Figures and Tables

**Figure 1 foods-11-01258-f001:**
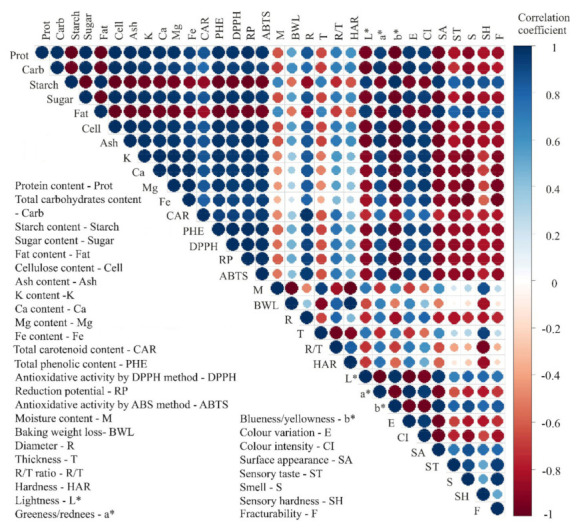
Colour correlation diagram between all 32 tested responses of cookies, with and without addition of L and OL peach.

**Figure 2 foods-11-01258-f002:**
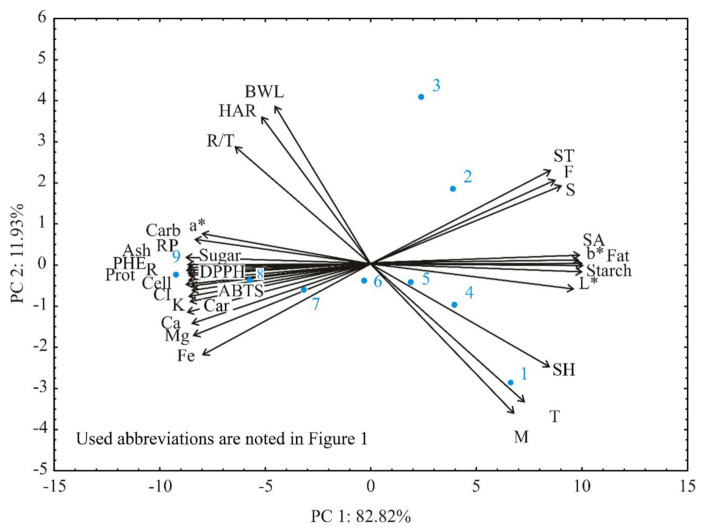
PCA of tested cookies with and without the addition of L and OL peach.

**Table 1 foods-11-01258-t001:** Experimental plan and formulation for cookie samples with and without the addition of lyophilized and osmodehydrated and lyophilized peach.

Sample No.	1	2	3	4	5	6	7	8	9
Quantity of added dehydrated peach (% dry matter on flour dry matter)	0	2.50	5.00	2.50	5.00	10.00	15.00	20.00	25.00
White wheat flour (g dry matter)	193.50	188.78	184.28	188.78	184.28	175.91	168.26	161.25	154.8
Margarine (g)	64.00	64.00	64.00	64.00	64.00	64.00	64.00	64.00	64.00
Sugar (g)	152.50	152.50	152.50	152.50	152.50	152.50	152.50	152.50	152.50
NaCl (g)	2.10	2.10	2.10	2.10	2.10	2.10	2.10	2.10	2.10
NaHCO_3_ (g)	2.50	2.50	2.50	2.50	2.50	2.50	2.50	2.50	2.50
Glucose (g)	2.00	2.00	2.00	2.00	2.00	2.00	2.00	2.00	2.00
Distilled water (g)	49.00	30.12	12.12	46.98	45.05	41.46	38.18	35.18	32.41
Lyophilized (L) peach (g)	0	23.6	46.1	0	0	0	0	0	0
Osmodehydrated and lyophilized (OL) peach (g)	0	0	0	6.74	13.17	25.13	36.06	46.07	55.29

**Table 2 foods-11-01258-t002:** Chemical and mineral matter content of cookies with and without addition of L and OL peach.

Sample no.	Content of										
	Protein	Total Carbohydrates	Starch	Sugar	Fat	Cell	Ash	K	Ca	Mg	Fe
	(% d.m.)	(% d.m.)	(% d.m.)	(% d.m.)	(% d.m.)	(% d.m.)	(% d.m.)	(mg/100 g)	(mg/100 g)	(mg/100 g)	(mg/100 g)
1	5.37 ±	69.49 ±	32.35 ±	37.22 ±	10.97 ±	7.62 ±	0.35 ±	125.73 ±	18.27 ±	25.43 ±	1.31 ±
	0.04 ^a^	0.87 ^a^	0.23 ^a^	0.60 ^a^	0.17 ^a^	0.03 ^b^	0.00 ^a^	1.59 ^a^	0.11 ^b^	0.28 ^ab^	0.02 ^ab^
2	5.41 ±	69.68 ±	31.97 ±	37.70 ±	10.84 ±	7.81 ±	0.42 ±	125.7 ±	17.79 ±	24.71 ±	1.28 ±
	0.05 ^ab^	0.33 ^a^	0.33 ^a^	0.20 ^a^	0.06 ^a^	0.08 ^ab^	0.00 ^b^	0.47 ^a^	0.07 ^ab^	0.28 ^a^	0.01 ^ac^
3	5.44 ±	69.74 ±	31.58 ±	38.16 ±	10.72 ±	7.93 ±	0.45 ±	126.38 ±	17.61 ±	24.48 ±	1.26 ±
	0.10 ^ab^	0.98 ^a^	0.22 ^ab^	0.42 ^ab^	0.12 ^ab^	0.16 ^ac^	0.01 ^c^	1.24 ^a^	0.22 ^a^	0.33 ^a^	0.02 ^c^
4	5.42 ±	69.67 ±	32.05 ±	37.67 ±	10.85 ±	7.80 ±	0.43 ±	136.29 ±	19.29 ±	26.47 ±	1.31 ±
	0.06 ^ab^	0.76 ^a^	0.12 ^a^	0.38 ^a^	0.10 ^a^	0.07 ^ab^	0.00 ^b^	2.61 ^b^	0.19 ^c^	0.30 ^b^	0.01 ^ab^
5	5.45 ±	69.73 ±	31.62 ±	38.13 ±	10.70 ±	7.94 ±	0.46 ±	146.86 ±	20.60 ±	28.19 ±	1.34 ±
	0.05 ^ab^	0.34 ^a^	0.21 ^ab^	0.16 ^ab^	0.14 ^ab^	0.13 ^ac^	0.01 ^c^	1.12 ^c^	0.31 ^d^	0.42 ^c^	0.01 ^b^
6	5.51 ±	69.88 ±	31.09 ±	38.92 ±	10.47 ±	8.22 ±	0.55 ±	166.97 ±	22.79 ±	30.79 ±	1.39 ±
	0.04 ^a–c^	0.72 ^a^	0.36 ^be^	0.38 ^bc^	0.12 ^be^	0.06 ^cd^	0.00 ^d^	2.08 ^d^	0.28 ^e^	0.29 ^d^	0.03 ^d^
7	5.58 ±	70.05 ±	30.45 ±	39.65 ±	10.22 ±	8.47 ±	0.61 ±	185.49 ±	24.61 ±	33.37 ±	1.41 ±
	0.06 ^b–d^	0.60 ^a^	0.07 ^de^	0.35 ^cd^	0.11 ^de^	0.10 ^de^	0.01 ^e^	1.86 ^e^	0.38 ^f^	0.60 ^e^	0.02 ^d^
8	5.62 ±	70.08 ±	29.95 ±	40.12 ±	10.09 ±	8.69 ±	0.68 ±	201.18 ±	26.74 ±	35.64 ±	1.45 ±
	0.06 ^cd^	0.43 ^a^	0.17 ^cd^	0.46 ^de^	0.09 ^cd^	0.053 ^ef^	0.01 ^f^	2.49 ^f^	0.10 ^g^	0.41 ^f^	0.03 ^e^
9	5.67 ±	70.16 ±	29.35 ±	40.74 ±	9.91 ±	8.83	0.74 ±	217.24 ±	28.91 ±	37.84 ±	1.48 ±
	0.05 ^d^	0.70 ^a^	0.31 ^c^	0.77 ^e^	0.07 ^c^	0.09 ^f^	0.00 ^g^	1.25 ^g^	0.21 ^h^	0.18 ^g^	0.01 ^e^

Results represent average value (number of repetitions (*n*) = 3) ± standard deviation ^a–h^. Different letters in superscript of the same table column indicate a statistically significant difference between values, at level of significance of *p* < 0.05 (based on post-hoc Tukey HSD test).

**Table 3 foods-11-01258-t003:** Content of total carotenoid and polyphenol compounds and antioxidative activity of cookies with and without the addition of L and OL peach.

	Sample No.								
	1	2	3	4	5	6	7	8	9
Total carotenoid content mg β-carE/100 g	nd *	nd	nd	nd	nd	0.02 ±	0.05 ±	0.17 ±	0.36 ±
						0.00 ^a^	0.00 ^b^	0.01 ^c^	0.02 ^d^
Total phenolic conent mg GAE/100 g	6.56 ±	9.31 ±	12.24 ±	10.19 ±	13.06 ±	17.85 ±	27.18 ±	30.62 ±	43.38 ±
	0.24 ^a^	0.32 ^b^	0.41 ^c^	0.27 ^b^	0.32 ^c^	0.18 ^d^	0.07 ^e^	1.31 ^f^	0.54 ^g^
Antioxidative activity by DPPH method μmol TE/100 g	0.08 ±	0.15 ±	0.18 ±	0.18 ±	0.22 ±	0.25 ±	0.62 ±	0.71 ±	1.00 ±
	0.00 ^a^	0.00 ^b^	0.01 ^bc^	0.02 ^bc^	0.00 ^cd^	0.01 ^d^	0.02 ^e^	0.04 ^f^	0.05 ^g^
Reduction potential μmol TE/100 g	19.28 ±	29.31 ±	40.85 ±	32.75 ±	42.29 ±	54.16 ±	90.23 ±	105.98 ±	145.83 ±
	1.07 ^a^	0.50 ^b^	0.77 ^c^	0.61 ^b^	0.74 ^c^	1.54 ^d^	2.08 ^e^	1.82 ^f^	3.04 ^g^
Antioxidative activity by ABTS method μmol TE/100 g	2.53 ±	7.42 ±	11.95 ±	15.46 ±	27.32 ±	42.83 ±	59.20 ±	89.98 ±	140.14 ±
	0.01 ^a^	0.08 ^b^	0.63 ^c^	0.08 ^c^	2.71 ^d^	0.85 ^e^	0.54 ^f^	1.87 ^g^	1.62 ^h^

* nd = not detected. Results represent average value (*n* = 6) ± standard deviation (for Table 3, Table 4 and Table 5) ^a–i^ Different letters in superscript of the same table row (for Table 3, Table 4, Table 5 and Table 6) indicate the statistically significant difference between values, at a level of significance of *p* < 0.05 (based on post-hoc Tukey HSD test).

**Table 4 foods-11-01258-t004:** Technological quality responses of cookies with and without the addition of L and OL peach.

	Sample No.								
	1	2	3	4	5	6	7	8	9
Moisture content (%)	12.14 ±	3.98 ±	2.82 ±	9.51 ±	6.76 ±	5.98 ±	4.89 ±	3.54 ±	2.95 ±
	0.10 ^h^	0.02 ^c^	0.02 ^a^	0.13 ^g^	0.13 ^f^	0.06 ^e^	0.03 ^d^	0.03 ^b^	0.01 ^a^
Baking weight loss (%)	6.08 ±	15.82 ±	18.18 ±	10.34 ±	12.11 ±	13.45 ±	14.86 ±	15.50 ±	16.34 ±
	0.03 ^a^	0.17 ^fg^	0.18 ^h^	0.08 ^b^	0.11 ^c^	0.13 ^d^	0.15 ^e^	0.18 ^f^	0.28 ^g^
Diameter (mm)	73.73 ±	72.92 ±	74.17 ±	73.09 ±	73.59 ±	73.92 ±	74.43 ±	75.75 ±	78.58 ±
	1.39 ^ab^	0.83 ^a^	0.19 ^ab^	0.93 ^a^	0.60 ^ab^	0.99 ^ab^	1.38 ^ab^	0.76 ^b^	0.29 ^c^
Thickness (mm)	14.09 ±	11.09 ±	9.13 ±	12.38 ±	11.86 ±	11.69 ±	11.43 ±	10.05 ±	8.68 ±
	0.16 ^g^	0.11 ^d^	0.13 ^b^	0.12 ^f^	0.04 ^e^	0.12 ^e^	0.13 ^de^	0.13 ^c^	0.18 ^a^
Diameter to thickness ratio (R/T)	5.23 ±	6.58 ±	8.12 ±	5.90 ±	6.20 ±	6.32 ±	6.51 ±	7.54 ±	9.05 ±
	0.02 ^a^	0.12 ^e^	0.06 ^g^	0.05 ^b^	0.07 ^c^	0.13 ^cd^	0.06 ^de^	0.10 ^f^	0.02 ^h^
Hardness (*n*)	5.42 ±	43.85 ±	52.41 ±	21.29 ±	30.67 ±	38.17 ±	41.17 ±	45.41 ±	48.60 ±
	0.02 ^a^	0.36 ^f^	0.32 ^i^	0.17 ^b^	0.61 ^c^	0.28 ^d^	0.60 ^e^	0.43 ^g^	0.82 ^h^

**Table 5 foods-11-01258-t005:** Instrumental colour responses of cookies with and without the addition of L and OL peach.

	Sample No.								
	1	2	3	4	5	6	7	8	9
L*	57.02 ±	48.02 ±	44.28 ±	45.37 ±	42.48 ±	39.65 ±	36.06 ±	33.73 ±	29.92 ±
	0.6 ^h^	0.72 ^g^	0.22 ^f^	0.47 ^f^	0.23 ^e^	0.32 ^d^	0.54 ^c^	0.3 ^b^	0.37 ^a^
a*	8.50 ±	10.62 ±	11.43 ±	10.92 ±	11.98 ±	12.42 ±	12.88 ±	13.6 ±	14.45 ±
	0.0 ^a^	0.15 ^b^	0.13 ^c^	0.11 ^b^	0.12 ^d^	0.18 ^e^	0.14 ^f^	0.12 ^g^	0.23 ^h^
b*	22.14 ±	20.70 ±	19.66 ±	19.35 ±	18.24 ±	17.51 ±	16.10 ±	14.53 ±	12.74 ±
	0.39 ^g^	0.24 ^f^	0.14 ^e^	0.20 ^e^	0.33 ^d^	0.10 ^d^	0.32 ^c^	0.14 ^b^	0.12 ^a^
ΔΕ	-	9.36 ±	13.31 ±	12.22 ±	15.45 ±	18.40 ±	22.25 ±	25.04 ±	29.29 ±
		0.05 ^a^	0.07 ^c^	0.05 ^b^	0.14 ^d^	0.11 ^e^	0.25 ^f^	0.40 ^g^	0.30 ^h^

**Table 6 foods-11-01258-t006:** Descriptive sensory analysis of cookies with and without the addition of L and OL peach.

	Sample No.								
	1	2	3	4	5	6	7	8	9
Colour intensity	4.00 ±	4.10 ±	4.60 ±	4.70 ±	5.60 ±	5.70 ±	5.9 ±	6.30 ±	6.90 ±
	0.20 ^a^	0.30 ^a^	0.20 ^a^	0.10 ^a^	0.40 ^b^	0.30 ^b^	0.12 ^b^	0.40 ^bc^	0.20 ^c^
Surface appearance	6.80 ±	6.60 ±	5.00 ±	5.80 ±	4.40 ±	3.80 ±	2.90 ±	2.40 ±	1.90 ±
	0.20 ^f^	0.40 ^f^	0.10 ^d^	0.20 ^e^	0.20 ^cd^	0.10 ^c^	0.20 ^b^	0.10 ^ab^	0.30 ^a^
Taste	4.00 ±	4.60 ±	5.60 ±	4.10 ±	4.30 ±	4.40 ±	3.40 ±	2.30 ±	1.80 ±
	0.00 ^bc^	0.40 ^cd^	0.50 ^d^	0.40 ^bc^	0.30 ^bc^	0.30 ^bc^	0.40 ^b^	0.40 ^a^	0.20 ^a^
Smell	4.00 ±	4.40 ±	4.80 ±	4.20 ±	3.80 ±	3.20 ±	2.70 ±	2.50 ±	2.30 ±
	0.00 ^c–e^	0.60 ^de^	0.30 ^e^	0.20 ^de^	0.30 ^cd^	0.40 ^bc^	0.10 ^ab^	0.20 ^ab^	0.20 ^a^
Hardness	4.00 ±	3.30 ±	2.70 ±	3.90 ±	3.70 ±	3.40 ±	3.00 ±	2.50 ±	2.20 ±
	0.00 ^f^	0.20 ^c–e^	0.30 ^a–c^	0.10 ^ef^	0.40 ^ef^	0.20 ^d–f^	0.10 ^b–d^	0.30 ^ab^	0.10 ^a^
Fractur-ability	4.00 ±	4.20 ±	5.40 ±	4.50 ±	3.80 ±	3.50 ±	3.00 ±	2.40 ±	2.00 ±
	0.00 ^ab^	0.10 ^ab^	0.20 ^b^	0.30 ^ab^	1.64 ^ab^	1.50 ^ab^	1.48 ^ab^	0.20 ^a^	0.10 ^a^

Results represent average value (*n* = 10) ± standard deviation.

**Table 7 foods-11-01258-t007:** Z-score analysis of cookies with and without the addition of L and OL peach.

Sample No.	Segment Z-Score						Total Z-Score
	S_1_	S_2_	S_3_	S_4_	S_5_	S_6_	S
1	0.00	0.09	0.00	0.98	0.75	0.81	0.54
2	0.16	0.03	0.05	0.43	0.64	0.79	0.38
3	0.27	0.00	0.10	0.18	0.58	0.78	0.31
4	0.16	0.16	0.08	0.75	0.57	0.77	0.48
5	0.28	0.28	0.14	0.60	0.53	0.60	0.45
6	0.49	0.49	0.22	0.52	0.47	0.49	0.47
7	0.71	0.65	0.45	0.44	0.39	0.31	0.49
8	0.84	0.83	0.63	0.27	0.34	0.13	0.46
9	1.00	1.00	1.00	0.04	0.25	0.00	0.45

## Data Availability

Data is contained within the article or Appendix A.

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
