# Peer review of "Addition of Combinedly Dehydrated Peach to the Cookies—Technological Quality Testing and Optimization"

_foods, 2022, doi:10.3390/foods11091258_

Round 1

Reviewer 1 Report

The organization of  this manuscript is very poor, the authors needs to reorganize the whole manuscript. 

  1. When peach was mixed with flour to make cookie, the appearance of peach needs to be describe in details. i.e. it is in powder or small piece, and how big is the pieces approximately.  Also different peach varieties are very different, the authors should not only state they are from grocery store. 
  2. The whole results part needs to be rewritten, the authors should check other scientific papers. The results does not mean you only to explain all the figures and tables. Also the legend of the table and figures dont need to be state again. 
  3. The discussions part also needs to be rewritten and check other papers. This part needs to be divided into sub-sections with subtitles. There should be an conclusion paragraph to summarize the whole study
  4.  this is a break in line 629-630

Author Response

Reviewer 1

Open Review

English language and style

(x) Extensive editing of English language and style required
( ) Moderate English changes required
( ) English language and style are fine/minor spell check required
( ) I don't feel qualified to judge about the English language and style

Yes

Can be improved

Must be improved

Not applicable

Does the introduction provide sufficient background and include all relevant references?

( )

(x)

( )

( )

Is the research design appropriate?

( )

(x)

( )

( )

Are the methods adequately described?

( )

(x)

( )

( )

Are the results clearly presented?

( )

( )

(x)

( )

Are the conclusions supported by the results?

( )

( )

(x)

( )

AUTHORS:

The authors would like to thank the Reviewer for professional and helpful comments. It is obvious that the Reviewer is an expert in this field. The Reviewer`s comments contribute to better quality of the paper that was submitted. All remarks are accepted and paper is changed according to these comments.

English language is checked and corrected by native English language speaker.

Introduction section is supplemented with references regarding peach organoleptic and nutritive description. The prupose of incorporating dehydrated peach in cookies formulation is also stated.

Reviewer 1:

Comments and Suggestions for Authors

The organization of  this manuscript is very poor, the authors needs to reorganize the whole manuscript. 

AUTHORS:

Manuscript organization is completly changed. Results and discussion sections are merged and rewriten. Conclusion section is created, and postitioned after Results and discussion section, where summarized conclusions of the research are presented.

Reviewer 1:

When peach was mixed with flour to make cookie, the appearance of peach needs to be describe in details. i.e. it is in powder or small piece, and how big is the pieces approximately.  Also different peach varieties are very different, the authors should not only state they are from grocery store. 

AUTHORS:

Cultivar of the peaches is added to the 2.1 Material section:

Fresh peaches (Prunus persica, var. nucipersica, Big top cultivar)

Description of peach samples preparation was added to the 2.4. Cookie samples production section:

Peach samples were cut into small pieces of approximate dimenssions of: 0.5x0.5x0.5 cm, in effort of better homogenization with the rest of the dough components.

Reviewer 1:

The whole results part needs to be rewritten, the authors should check other scientific papers. The results does not mean you only to explain all the figures and tables. Also the legend of the table and figures dont need to be state again. 

The discussions part also needs to be rewritten and check other papers. This part needs to be divided into sub-sections with subtitles. There should be an conclusion paragraph to summarize the whole study

AUTHORS:

Results and discussion section is made integral section, that is divided in logical subsections, analysing the same tested responses. In this manner repetition of some text sections was omited, and the text is more condensed and easier to follow. Changes in the text are too extensive to be presented in this document, they are marked in manuscript text.

Recurring same legends for different tables is omitted.

Disscussion of the presented results is broadend with additional comparison to the works of other authors.

Conclusion section is added, summarizing key findings of the presented research. Following text is added:

From presented results it can be concluded that different dehydration methods had statistically significantly different impact on nutritive characteristics of cookies, where molasses application in osmotic treatment stage of combined dehydration method had positive effect on cookies with dehydrated peaches overall nutritive composition. Cookies with 10% of combinedly dehydrated peach addition nutritive quality indicates on significant increase of chemical, mineral and phenol content and antioxidative activity. Peaches’ combined dehydration method positively influenced certain cookies’ technological quality parameters (BWL, R/T, hardness) in comparison to the cookies with the same amount of lyophilized peaches addition. The results of descriptive sensory analysis showed that with the addition of up to 10% of dehydrated peaches to the cookies’ formulation, there was positive effect on all sensory responses, providing favorable peach note to overall cookies’ tase and flavor. Correlation analysis results confirmed positive effect of dehydrated peach addition to the cookies formulation on nutritive characteristics, while technological quality parameters were mostly negatively correlated. The results of Z-score analysis indicated on optimal level of dehydrated peach addition, regarding overall cookies quality (quantity of 15% of addition), proposing a new type of nutritively enriched cookie product, as the main results of this research.

Reviewer 1:

this is a break in line 629-630

AUTHORS:

Corrected

Reviewer 2 Report

I have minor comments . Compared with other similar studies (e.g., Food Chemistry, 201, 129125), is there any difference between adding grains and fruits on the quality of product? It is suggested to talk about the effect on cookie digestion and aroma with the addition of dehydrated peaches. These questions could be explained in the discussion part.

Author Response

Reviewer 2

Open Review

English language and style

( ) Extensive editing of English language and style required
( ) Moderate English changes required
(x) English language and style are fine/minor spell check required
( ) I don't feel qualified to judge about the English language and style

Yes

Can be improved

Must be improved

Not applicable

Does the introduction provide sufficient background and include all relevant references?

( )

(x)

( )

( )

Is the research design appropriate?

(x)

( )

( )

( )

Are the methods adequately described?

(x)

( )

( )

( )

Are the results clearly presented?

(x)

( )

( )

( )

Are the conclusions supported by the results?

( )

(x)

( )

( )

Comments and Suggestions for Authors

AUTHORS:

The authors would like to thank the Reviewer for presented comments. The Reviewer`s comments contribute to better quality of the paper that was submitted. All remarks are accepted and paper is changed according to these comments.

Introduction section is supplemented with references regarding peach organoleptic and nutritive description. The prupose of incorporating dehydrated peach in cookies formulation is also stated.

Reviewer 2:

I have minor comments. Compared with other similar studies (e.g., Food Chemistry, 201, 129125), is there any difference between adding grains and fruits on the quality of product? It is suggested to talk about the effect on cookie digestion and aroma with the addition of dehydrated peaches. These questions could be explained in the discussion part.

AUTHORS:

Results and discussion section (new, integral section) is supplemented with comparison of the effect of wheat malt flour addition to cookies phenolic content, with dehydrated peach addition and its’ effect on phenolic content. Disscussion of the presented results is broadend with additional comparison to the works of other authors.

Cookies’ in vitro digestion analysis was not performed yet, although there are plans for further analysis in this research direction. Numerous other responses of nutritional and technological quality attributes (32 responses in total) are presented in this manuscript, hence it was decided, due to the volume of presented results, to limit addition of new results.

Cookies’ aroma components are analysed in detail via descriptive sensory analysis, dividing the effects of dehydrated peach addition on cookies on different colour, taste, smell and texture organoleptic properties, in effort of precisely detecting the optimal amount of dehydrated peach addition, from the aspect of cookies sensory acceptability.

Reviewer 3 Report

In this study, the authors incorporated dehydrated peach into standard sweet cookies to test and optimize the level of addition to cookie formulation. Dehydrated peach was prepared with two procedure; 1. Lyophilization and 2. Osmotic dehydration prior to lyophilization. Cookie samples were analyzed in terms of chemical composition (protein, total carbohydrates, starch, reducing sugar, fat, cellulose, ash); mineral matter composition (potassium (K), calcium (Ca), magnesium (Mg) and iron (Fe)); total carotenoids, polyphenol content and antioxidant activity; technological quality (Baking weight loss, cookie dimensions); textural properties; color and sensory properties.

In my opinion, the manuscript is appropriate for Foods. 

Here are my comments;

  • The authors should better indicate the purpose of peach addition to the cookie formulation. This is an important and missing point and should be clearly mentioned in the introduction part. The introduction part must be improved to provide sufficient background.
  • In the manuscript “table” and “figure” are written with lower case. The first letter should be upper case. For example; Page 3 Line; table 1 should be Table 1. Another example for Figure is figure at Line 639; it should be Figure instead of figure. Whole manuscript should be checked accordingly.
  • Two procedures were used to prepare dehydrated peach; 1. Lyophilization and 2. Osmotic dehydration prior to lyophilization. In the manuscript it is sometimes hard to follow the samples. In my opinion it would be better to code these procedures/samples. For example L for the lypohilized ones; and OL for the osmodehydrated and lyophilized ones.
  • In Table 4, R/T is used as abbreviated. The whole explanation of R/T should be written under the table.
  • There are some typo mistakes I saw, esspecially (Line 465) and moisture (Line 668). Please check the manuscript again.
  • I recommend to the authors to combine results and discussion part. There is an insufficient linkage of the results to the literature. A conclusion part (which is not found in this manuscript) may improve the manuscript.

Author Response

Reviewer 3

Open Review

English language and style

( ) Extensive editing of English language and style required
(x) Moderate English changes required
( ) English language and style are fine/minor spell check required
( ) I don't feel qualified to judge about the English language and style

Yes

Can be improved

Must be improved

Not applicable

Does the introduction provide sufficient background and include all relevant references?

( )

( )

(x)

( )

Is the research design appropriate?

(x)

( )

( )

( )

Are the methods adequately described?

(x)

( )

( )

( )

Are the results clearly presented?

( )

(x)

( )

( )

Are the conclusions supported by the results?

( )

( )

(x)

( )

Comments and Suggestions for Authors

In this study, the authors incorporated dehydrated peach into standard sweet cookies to test and optimize the level of addition to cookie formulation. Dehydrated peach was prepared with two procedure; 1. Lyophilization and 2. Osmotic dehydration prior to lyophilization. Cookie samples were analyzed in terms of chemical composition (protein, total carbohydrates, starch, reducing sugar, fat, cellulose, ash); mineral matter composition (potassium (K), calcium (Ca), magnesium (Mg) and iron (Fe)); total carotenoids, polyphenol content and antioxidant activity; technological quality (Baking weight loss, cookie dimensions); textural properties; color and sensory properties.

In my opinion, the manuscript is appropriate for Foods. 

AUTHORS:

The authors would like to thank the Reviewer for very detail comments. The Reviewer`s comments contribute to better quality of the paper that was submitted. All remarks are accepted and paper is changed according to these comments.

Here are my comments;

Reviewer 3:

  • The authors should better indicate the purpose of peach addition to the cookie formulation. This is an important and missing point and should be clearly mentioned in the introduction part. The introduction part must be improved to provide sufficient background.

AUTHORS:

Introduction section is supplemented with references regarding peach organoleptic and nutritive description. The prupose of incorporating dehydrated peach in cookies formulation is also stated.

Reviewer 3:

  • In the manuscript “table” and “figure” are written with lower case. The first letter should be upper case. For example; Page 3 Line; table 1 should be Table 1. Another example for Figure is figure at Line 639; it should be Figure instead of figure. Whole manuscript should be checked accordingly.

AUTHORS:

All „table“ and „figure“ words throught the manuscript text are changed in upper case letters.

Reviewer 3:

  • Two procedures were used to prepare dehydrated peach; 1. Lyophilization and 2. Osmotic dehydration prior to lyophilization. In the manuscript it is sometimes hard to follow the samples. In my opinion it would be better to code these procedures/samples. For example L for the lypohilized ones; and OL for the osmodehydrated and lyophilized ones.

AUTHORS:

Abbreviations of: L for lyophilization and OL for osmotic dehydration and succesive lyophilization are introduced in material and methods section, 2.4. Cookie samples production subsection, and are used throughtout the text.

Reviewer 3:

  • In Table 4, R/T is used as abbreviated. The whole explanation of R/T should be written under the table.

AUTHORS:

Followiing text is inserted to the table 4:

Diameter to thickness ratio (R/T)

Reviewer 3:

  • There are some typo mistakes I saw, esspecially (Line 465) and moisture (Line 668). Please check the manuscript again.

AUTHORS:

English language is checked and corrected by native English language speaker.

Reviewer 3:

  • I recommend to the authors to combine results and discussion part. There is an insufficient linkage of the results to the literature. A conclusion part (which is not found in this manuscript) may improve the manuscript.

AUTHORS:

Results and discussion section is made integral section, that is divided in logical subsections, analysing the same tested responses. In this manner repetition of some text sections was omited, and the text is more condensed and easier to follow.

Disscussion of the presented results is broadend with additional comparison to the works of other authors.

Changes in the text are too extensive to be presented in this document, they are marked in manuscript text.

Conclusion section is added, summarizing key findings of the presented research. Following text is added:

From presented results it can be concluded that different dehydration methods had statistically significantly different impact on nutritive characteristics of cookies, where molasses application in osmotic treatment stage of combined dehydration method had positive effect on cookies with dehydrated peaches overall nutritive composition. Cookies with 10% of combinedly dehydrated peach addition nutritive quality indicates on significant increase of chemical, mineral and phenol content and antioxidative activity. Peaches’ combined dehydration method positively influenced certain cookies’ technological quality parameters (BWL, R/T, hardness) in comparison to the cookies with the same amount of lyophilized peaches addition. The results of descriptive sensory analysis showed that with the addition of up to 10% of dehydrated peaches to the cookies’ formulation, there was positive effect on all sensory responses, providing favorable peach note to overall cookies’ tase and flavor. Correlation analysis results confirmed positive effect of dehydrated peach addition to the cookies formulation on nutritive characteristics, while technological quality parameters were mostly negatively correlated. The results of Z-score analysis indicated on optimal level of dehydrated peach addition, regarding overall cookies quality (quantity of 15% of addition), proposing a new type of nutritively enriched cookie product, as the main results of this research.

Round 2

Reviewer 1 Report

I am ok with the manuscript now.